# Functional Domains of the Herpes Simplex Virus Type 1 Tegument Protein pUL37: The Amino Terminus is Dispensable for Virus Replication in Tissue Culture

**DOI:** 10.3390/v11090853

**Published:** 2019-09-14

**Authors:** Peter Grzesik, Erin N. Pryce, Akshay Bhalala, Mannika Vij, Ray Ahmed, Lyns Etienne, Patric Perez, J. Michael McCaffery, Prashant J. Desai

**Affiliations:** 1Department of Oncology, The Sidney Kimmel Comprehensive Cancer Center at Johns Hopkins, The Johns Hopkins University, Baltimore, MD 21231, USA; pete4grz@gmail.com (P.G.); akshaybhalala@gmail.com (A.B.); mannikavij@gmail.com (M.V.); rayahmed1@gmail.com (R.A.); Lyns.Etienne@Agenusbio.com (L.E.); 2Integrated Imaging Center, Department of Biology, The Johns Hopkins University, Baltimore, MD 21218, USA; epryce@jhu.edu (E.N.P.); patricjperez@gmail.com (P.P.); jmccaffery@jhu.edu (J.M.M.)

**Keywords:** herpes simplex virus, tegument protein, pUL37, GFP, protein localization

## Abstract

The herpes simplex virus type 1 (HSV-1) UL37 gene encodes for a multifunctional component of the virion tegument, which is necessary for secondary envelopment in the cytoplasm of infected cells, for motility of the viral particle, and for the first steps in the initiation of virus infection. This 120 kDa protein has several known viral interacting partners, including pUL36, gK/pUL20, pUS10, and VP26, and cellular interacting proteins which include TRAF6, RIG-I, and dystonin. These interactions are likely important for the functions of pUL37 at both early and late stages of infection. We employed a genetic approach to determine essential domains and amino acid residues of pUL37 and their associated functions in cellular localization and virion morphogenesis. Using marker-rescue/marker-transfer methods, we generated a library of GFP-tagged pUL37 mutations in the HSV-1 strain KOS genome. Through viral growth and ultra-structural analysis, we discovered that the C-terminus is essential for replication. The N-terminal 480 amino acids are dispensable for replication in cell culture, although serve some non-essential function as viral titers are reduced in the presence of this truncation. Furthermore, the C-terminal 133 amino acids are important in so much that their absence leads to a lethal phenotype. We further probed the carboxy terminal half of pUL37 by alanine scanning mutagenesis of conserved residues among alphaherpesviruses. Mutant viruses were screened for the inability to form plaques—or greatly reduced plaque size—on Vero cells, of which 22 mutations were chosen for additional analysis. Viruses discovered to have the greatest reduction in viral titers on Vero cells were examined by electron microscopy (EM) and by confocal light microscopy for pUL37–EGFP cellular localization. This genetic approach identified both essential and non-essential domains and residues of the HSV-1 UL37 gene product. The mutations identified in this study are recognized as significant candidates for further analysis of the pUL37 function and may unveil previously undiscovered roles and interactions of this essential tegument gene.

## 1. Introduction

The assembly of virus particles has been used as a paradigm for how proteins interact and come together to form large multi-protein complexes. The herpesvirus virion is comprised of four structural components: An icosahedral capsid, which encloses the viral DNA genome; an electron-dense asymmetrically distributed material, which immediately surrounds the capsid and is termed the tegument; and an outer membrane or envelope, which encloses the tegument and capsid and in which are embedded the viral glycoproteins [1,2,3,4]. Capsid assembly, DNA packaging, nuclear exit, and cytoplasmic envelopment involve the participation of a large and diverse set of about 50 proteins. Herpesviruses, like other viruses, hijack the cell machinery for their propagation and morphogenesis. The nuclear lamina is disrupted to facilitate capsid access to the nuclear envelope, the cell cytoskeleton is used to transport capsids and sub-viral structures to sites of maturation in order to facilitate their egress, and the Golgi is modified to create budding sites for production of progeny virions, reviewed in [5,6,7,8,9,10,11,12,13,14,15,16,17,18]. For all herpesviruses, lytic replication serves a conduit to propagate the infection to naïve cells, therefore promoting latency in additional cellular reservoirs. These lytic replication pathways are important not only for virus propagation, but as mediators of immune evasion and cell specific replication.

The tegument is one of the most complex and diverse structures of the virion, both in terms of protein composition and the functions encoded by the constituents of this structure. The viral specified polypeptides that comprise this structure include those that function to activate transcription, shut off host protein synthesis, uncoat the virus genome, phosphorylate virus proteins, and others whose functions are still poorly defined, reviewed in [10,16,17,19,20]. The tegument displays a duality of functions in virus replication due to the role the proteins resident in this structure play, both at early and at late times in infection. What has become increasingly evident is the importance of the tegument proteins in the maturation process of the enveloped virus. To date, three tegument proteins resident in the mature HSV-1 virion have been shown to have a deleterious and complete lethal effect on the maturation process. These are VP16 [21,22], pUL36 (VP1/2) [23,24,25,26], and the product of the UL37 gene [24,27,28].

The UL37 gene specifies a 120 kD polypeptide [29] that is located in the tegument [30,31,32,33] phosphorylated [34] and localizes predominantly in the cytoplasm, with significant accumulation in the Golgi organelle [31,35]. It is, as stated, a component of the tegument, and more precisely, it is a constituent of the inner tegument [33,36]. This location is likely because of its physical interaction with the large tegument protein, pUL36 (VP1/2) that is capsid associated [3,37,38,39,40,41]. This complex of proteins plays an important role in transport of the mature capsid during both retrograde and anterograde movement in the infected epithelial cell and in the infected neuronal cell of the host [42,43,44,45,46]. In fact, recent studies show that pUL37 is an important pathogenic determinant in the mouse neuronal model, and a conserved motif called R2 in the N-terminus is important for neuroinvasion [47,48,49,50]. Many protein–protein interactions have been identified for pUL37 and the function of some of these interactions have been elucidated. The pUL36–pUL37 interaction is important for capsid association (together with VP26) [51,52,53,54,55,56,57], microtubule-based transport [42,43,44,47,58], Golgi localization [35], and virus maturation [25,37,59]. The pUL37–gK–pUL20 interaction is also an important determinant for cytoplasmic envelopment [60,61]. Recent studies also show pUL37–pUS10 interaction using a bioinformatics and biochemical approach, and this interaction may represent another pathway for virus envelopment, potentially involving gE [62]. pUL37 also interacts with host proteins; these include its activity as a deamidase of cellular RIG-I, thus preventing its activation and illustrating how pUL37 is an important factor for disarming the innate immune response to the virus. It also uses this deamidase to antagonize the cGAS–STING immune pathway in mice [63,64,65]. In addition, pUL37 also activates the NFκB pathway via binding to the NF-κB-signaling adaptor protein, TRAF6; this activity again highlighting a mechanism by which pUL37 usurps cellular pathways to facilitate virus replication [66]. Physical interactions between pUL37 and the spectraplakin protein, dystonin/BPAG1, has been reported. This interaction potentially links capsid-associated pUL37 with the microtubule transport system [45]. A map of the important functional and interaction domains of pUL37 is illustrated in Figure 1A.

Many years ago, our laboratory isolated a UL37 null mutant virus. The mutation was lethal for virus replication and although capsid assembly occurred normally, the cytoplasmic located nucleocapsids did not mature into enveloped infectious virus particles [27]. Additionally, Bucks et al. probed the functional domains of pUL37 using transient complementation assays. They discovered an important role for the carboxy terminal half of the protein, but could not analyze the defect in infected cells [67]. We decided to examine pUL37 functional domains and important residues in the context of the virus infected cell. What we discovered is that the N-terminal portion of pUL37 is dispensable for replication in tissue cultured epithelial cells, and there are important amino acids in the C-terminal portion of pUL37 that impart a lethal phenotype on virus replication.

## 2. Materials and Methods

### 2.1. Cells and Viruses

Vero cells, transformed Vero cell lines (BD45, BH31, and HS30), and retinal pigmented epithelial 1 (RPE-1) cells (telomerase immortalized) (ATCC) were all grown in minimal essential medium (alpha medium—Thermo Fisher, Waltham, MA, USA) supplemented with 10% fetal bovine serum (FBS—Thermo Fisher) and passaged as described previously [68]. All stocks of HSV-1 viruses were amplified as also described by Desai et al. [68], except in some cases mutant viruses were not amplified for working stocks but were used for analyses from the secondary stock preparations.

### 2.2. DNA Plasmids

Plasmid pKBD, which encodes a genomic fragment of HSV-1 strain KOS (Accession: JQ673480) [69] in which resides the UL37 gene (genome nucleotide number: 80,641–84,012), has been previously described [27]. The fragment spans from a BamHI site (79,376) to a DraI restriction enzyme site (84,293), Figure 1C. Subsequently, the EGFP ORF was inserted into the SpeI site at 80,650 at the C-terminus of UL37 (pKUL37eGFP) [35]. We also generated another plasmid that contains a larger genomic region surrounding UL37. This was made by cloning an EcoRI (74,866) to NheI (85,225) fragment derived from plasmid pKBGD (cloned BglIID fragment of HSV-1 KOS [23]) and cloned into the EcoRI/XbaI sites of pUC19. Into this plasmid, named pKNE37, we similarly inserted the EGFP ORF at the unique SpeI site in correct orientation to derive plasmid pKNE37eGFP (Figure 1C). This plasmid serves as a basis for the introduction of all the UL37 truncation mutants prior to recombination into the virus genome. To facilitate this, we also engineered a BglII restriction enzyme site in pKNE37eGFP after the start of the UL37 gene (84,009) and a “silent” BsrGI restriction enzyme site approximately in the middle of the gene at 82,555 using QuikChange methods (see below).

The different UL37 truncations were made using primers that amplified the designed sequences (Figure 1B). The forward primer contained an EcoRI site, and the reverse primer, a BamHI site. Thus all inserts were then cloned as EcoRI–BamHI fragments into the same sites of pEGFP-N2 (Clontech, Mountain View, CA, USA), thus generating UL37 mutants with a C-terminal fusion to EGFP. Following sequence analysis to confirm the correct amplified gene fragment, these DNAs were used subsequently as templates to amplify the different truncations for cloning into pKNE37eGFP, either as BglII–BsrG1 fragments (N-terminal) or BglII–SpeI fragments (C-terminal). Wild-type versions of the UL37 gene were also made using the same cloning methods. These plasmids were also sequenced using Sanger sequencing.

The triple alanine (AAA) scanning mutations were made using QuikChange protocols, as previously described by Walters et al. [70]. The template for this mutational experiment was plasmid pKUL37eGFP. Positive DNA clones carrying the mutation were confirmed by restriction enzyme (introduction of new NotI restriction enzyme site) and sequence analysis.

A plasmid that carries a deletion that spans the majority of UL37 and extends partway into UL36 was made in the pKBD plasmid background. A CMV-DsRed2 reporter cassette, derived from pDsRed2-N1 (Clontech), was inserted at the site of the deletion. To do this, first the multiple cloning sequence (MCS) in pDsRed2-N1 was deleted by digestion with BglII and BamHI, followed by re-ligation. The CMV-DsRed2 cassette was amplified using this pDsRed2-N1 ∆MCS plasmid as a template, and cloned as a SpeI–NotI fragment into pKBD that had been digested with AvrII (79,931) and NotI (83,755), Figure 1C. This plasmid was designated pK∆36/37CR and was used to generate a virus with a UL36 and UL37 deletion for the marker-rescue/marker-transfer experiments.

All the primers used and their sequences are shown in Appendix A.

### 2.3. Generation of Complementing Vero Cell Lines

To produce a cell line that complements both UL37 and UL36 mutants, we made a complementing cell line using methods described by Deluca et al. [72] and Desai et al. [73]. Low passage Vero cells were transformed with pSV2neo [74] and plasmids that carry genomic fragments in which reside the UL37 gene (pKBD [27]) and the UL36 gene (pKUL36 [23]). Complementing cells after clonal selection in the presence of G418 were selected for their ability to support the replication of both the UL37 null mutant virus, K∆UL37 [27], and the UL36 null mutant virus, K∆UL36 [23]. Out of 34 G418 resistant cell lines, we found three that supported the plaquing ability of both K∆UL37 and K∆UL36. One cell line, designated BH31, was chosen for our marker-rescue/marker-transfer method.

### 2.4. Isolation of K∆36/37 Virus Genome for Marker-Rescue/Marker-Transfer

To produce the recipient virus genome for the marker-rescue/marker-transfer experiments, the plasmid pK∆36/37CR was linearized using BamHI enzyme, and this linear molecule (1–2 µg) was co-transfected into BH31 cells (1 × 10^6^), together with 25 µL of KOS-infected cell DNA. Following 3 days incubation, the cells were harvested, freeze-thawed, and sonicated. Serial dilutions of the transfection lysate were plated on BH31 cell monolayers and, following plaque formation, visualized using a fluorescence microscope. Several plaques that displayed red fluorescence were plaque purified using limiting dilution prior to further characterization. Secondary stocks of the purified viruses were tested for their plating efficiency on Vero, BD45 (UL37 complementing) [27], and BH31 cell lines. Almost all formed plaques on BH31 cells but not on Vero or BD45 cell lines, thus confirming the phenotype of the ∆36/37 deletion. One isolate, now designated K∆36/37CR, was used for the marker-rescue/marker-transfer experiments and its genotype was confirmed by Southern blot analysis of the viral genome and compared to the KOS wild-type genome (data not shown).

### 2.5. Marker-Transfer and Marker-Rescue

Marker-transfer of the UL37 mutations was essentially carried out as described by Person and Desai [75]. Subconfluent monolayers of BH31 cells (1 × 10^6^) in 60-mm-diameter dishes were co-transfected with 2 µg of linearized plasmid (EcoRI) and 20–30 µL of K∆36/37CR infected cell DNA. When foci were observed (72 h after transfection), the cell monolayers were harvested, freeze- thawed once, and sonicated, and serial dilutions of the virus progeny were plated on BH31, BD45, and Vero cell monolayers to determine the lethality of the mutation (see Appendix A). The transfection progeny was subjected to plaque purification (3 rounds) on BD45 cells prior to further characterization. Secondary stocks were made by infecting BD45 cells (5 × 10^5^) with the final purified plaque. High titer working stocks were again made using BD45 cells following infection at a multiplicity of infection (MOI) of 0.05 plaque forming units per cell (PFU/cell) with the secondary stock. All viruses were analyzed by Sanger sequencing to confirm the correct deletion or amino acid substitution. To do this, BD45 cells (2.5 × 10^5^) were infected with the secondary stock and the infected cells harvested when all the cells displayed the typical cytopathic effects. The cell pellet was resuspended in water and a portion of this was used as the template for PCR using Phire polymerase (NEB, Ipswich, MA, USA and Thermo Fisher). The correct PCR product was gel extracted and sent for Sanger sequencing (Macrogen Inc., Rockville, MD, USA).

### 2.6. Western Blot Analysis of Infected Cell Lysates

Vero cells (5 × 10^5^) were infected at an MOI of 10 PFU/cell and harvested 24 h post-infection. Cell pellets were lysed in 2× Laemmli buffer and 10% of this sample was resolved using NuPAGE 4–12% Bis-Tris gels (Thermo Fisher) and transferred to nitrocellulose membranes using the iBlot2 system (Thermo Fisher), as described by Luitweiler et al. [76]. Rabbit antibody to GFP (Invitrogen A-6455, Thermo Fisher) was used at a dilution of 1:5,000. Blots were processed using the enhanced chemiluminescence (ECL) kit (GE Healthcare, Chicago, IL, USA) according the manufacturer’s protocol.

### 2.7. Fluorescence Light Microscopy Imaging

For confocal imaging, RPE-1 cells (5 × 10^5^) were seeded in a 4-well borosilicate glass bottom chamber slide (Lab-Tek, Nalgene Nunc, Rochester, NY, USA). Cells were infected with each virus at an MOI of 10 PFU/cell and overlaid with FluroBrite DMEM (Thermo Fisher) supplemented with 1% FetalPlex (Gemini Bioproducts, West Sacramento, CA, USA). Twelve hours after infection, cells were imaged on a Zeiss LSM 510 confocal microscope using 63× objective.

### 2.8. Transmission Electron Microscopy

Vero cells (8.6 × 10^6^ cells) in 100 mm tissue culture dishes were infected at an MOI of 10 PFU/cell and processed for transmission electron microscopy (TEM) experiments [27]. Infected cells were processed 18 h post-infection. Samples were examined using an FEI Tecnai 12 electron microscope; images were obtained with an SIS Megaview III camera (Olympus, Tokyo, Japan).

## 3. Results

### 3.1. Identification of Functional Domains of pUL37

Previous studies in our laboratory demonstrated the trafficking of pUL37 to the Golgi structure in infected cells. This was visualized in living cells using an enhanced green fluorescent protein (EGFP) tag on the protein [35]. This trafficking was dependent on the presence of functional pUL36, but capsids were not required. Although the predominant localization of pUL37, as judged by the GFP signal, was located in the Golgi, there was an additional fluorescence signal that was distributed diffusely throughout the cytoplasm. This fluorescence was absent from the nucleus. Our next step was to delineate domains of this protein that are required for this cytoplasmic trafficking in infected cells. We initially generated N-terminal and C-terminal truncations of UL37 fused to the EGFP ORF in the plasmid pEGFP-N2 and used these plasmids in transient transfection assays. The N-terminal truncations made deleted the first 39 (∆40N), 71 (∆72N), 249 (∆250N), 289 (∆290N), 371 (∆372N), and 480 (∆481N) amino acids. The C-terminal truncations were made such that the terminal 133 (∆990C), 393 (∆730C), 622 (∆501C), and 873 (∆250C) amino acids were deleted (Figure 1B). However, due to the strong expression of the polypeptide from the CMV promoter, we could not ascertain correctly the localization of the protein because of potential aggregation and cell toxicity issues. Thus, we changed our objective and cloned the different truncation polypeptides into a plasmid containing the UL37 genomic locus, and transferred these mutant genes into the KOS virus genome to analyze pUL37 trafficking in the infected cell.

### 3.2. Marker-Rescue/Marker-Transfer Genetic Method for Transfer of UL37 Mutations into the Virus Genome

To do this, we first established a marker-rescue/marker-transfer method for the introduction of any UL37 gene mutation into the KOS virus genome. Similar to the methods we developed previously [77], this requires two genes adjacent to each other to be essential for virus replication and cell lines that are transformed such that they can complement a double mutation in the two genes and in the gene of interest. The side-by-side arrangement of genes UL36 and UL37 (Figure 1C), and the fact that they are essential for virus replication, thus makes this method tractable. We first transformed low passage Vero cells with plasmids encoding pUL36, and also pUL37 and plasmid pSV2Neo, and selected for G418 resistant clones. These clones were screened for their ability to complement both K∆UL36 and K∆UL37, viruses that encode null mutations in the UL36 and UL37 genes individually [23,27]. We identified cell lines that were able to demonstrate this phenotype and chose one, designated BH31, for further analysis and use in the marker-rescue/marker-transfer method. The second requirement is to create a mutant virus in which a deletion is engineered, which includes the majority of the UL37 gene and part of the UL36 gene. This double mutant would replicate on BH31 cells but not on BD45 cells, the UL37 complementing cell line [27]. We generated such a mutant by deleting amino acids 1–919 of pUL37 (total length = 1123 amino acids) and the promoter element and the N-terminal 181 amino acids of pUL36. In the site of this deletion, we also introduced a reporter cassette that drives DsRed2 gene expression, which allowed us to select for this virus using red fluorescence visualized in the plaque (Figure 1C). The virus, following isolation and purification on BH31 cells and Southern blot analysis (data not shown), was designated K∆36/37CR and the phenotype this virus displayed was the ability to plaque on BH31 cells, but not on BD45 cells, nor on the pUL36 complementing cell line, HS30 [23] (Table 1).

### 3.3. Phenotypes of the UL37 Mutations Following Transfer into the Virus Genome

We used this virus as the recipient genome for the marker-rescue/marker-transfer of UL37 mutations. The truncation mutations were engineered in a modified plasmid, pKNE37, that contains an EGFP sequence at the unique SpeI restriction enzyme site at the C-terminus of UL37. Co-transfection of linearized plasmid and infected cell K∆36/37CR DNA into BH31 cells was carried out and the virus progeny harvested after 3 days post-transfection. We used the wild-type plasmid first, to demonstrate successful marker-rescue/marker-transfer. The progeny virus was able to replicate on Vero cells as expected and displayed GFP expression with cellular distribution similar to that seen previously. Virus progeny from transfections using the different truncation mutants gave mixed results. N-terminal truncations mutants ∆72N, ∆250N, ∆290N, and ∆372N failed to produce a virus that could replicate on Vero cells. All these mutants required BD45 cells for productive replication. This was also true for the C-terminal truncation mutants ∆250C, ∆501C, ∆730C, and ∆990C. Deletion of the amino terminal 39 amino acids (∆40N) of pUL37 did not affect virus replication on Vero cells. However, interestingly, a mutant virus (∆481N) that has a deletion of the N-terminal 480 amino acids was still able to replicate on Vero cells, albeit not to the levels observed with wild-type virus. An example of the plating efficiency of three different recombinants (WT [UL37–EGFP], ∆40N, and ∆501C) on the three cells lines (BH31, BD45, and Vero) is shown in Appendix A. All these viruses were purified further on BD45 or Vero cells, and the growth properties of the purified viruses examined using single-step growth analyses (Figure 2A,B). This quantitatively showed that virus ∆481N can replicate in Vero cells with a mean virus yield of nearly one fifth of wild-type virus. All the other mutants that specify lethal phenotypes could not produce virus progeny in Vero cells. All mutant viruses were also examined for pUL37 polypeptide expression in Vero cells using immunoblot methods (Figure 2C). Proteins of the correct molecular weight size were observed in the lysates of the different mutant virus-infected cells and most appeared to show stable accumulation.

### 3.4. Visualization of the Defects in Virus Maturation and pUL37 Localization

The defect in the ability of virus replication was investigated further using ultrastructural and light microscopy methods. First, Vero cells were infected with the different mutants and the cells processed for TEM following 18 h of infection (Figure 3). In wild-type infected cells, intranuclear capsids, cytoplasmic capsids, as well as enveloped virions in the cytoplasm and at the plasma membrane were readily observed. Enveloped particles were evident in cells infected with the replication competent viruses ∆40N and ∆481N (Figure 3A). For the remainder of mutant UL37 infected cells, there was no evidence of mature membrane bound virions in the many sections analyzed. Representative images for ∆290N and ∆990C are shown (Figure 3B), and all other viruses specifying lethal phenotypes in Vero cells were examined with no evidence of cytoplasmic virus particle envelopment (Appendix A).

Confocal microscopy was used to follow the cellular localization of pUL37 by virtue of the GFP tag on the polypeptide (Figure 4). RPE-1 cell monolayers were infected with each KUL37 mutant virus and imaged 12 h post-infection. Wild-type pUL37 and ∆40N trafficked to the Golgi complex, as evidenced by the juxtanuclear localization and previous data from Desai et al. [35]. N-terminal truncations of 71aa or more resulted in the dispersal of pUL37–EGFP localization to a more perinuclear localization. This included the functional Δ481N mutation (Figure 4A). C-terminal truncations all abolished the juxtanuclear localization pattern and may have some effects on the pUL37–EGFP cytoplasmic diffuse localization (Figure 4B).

### 3.5. Investigation of the Essential C-Terminal Domain of pUL37 Using Alanine Scanning Mutagenesis

Because it was previously discovered that the C-terminus of pUL37 specifies the most essential functions [67], which was confirmed by our data, we chose to probe this region using more refined site directed mutations. In this case we engineered triple alanine (AAA) substitutions that target different contiguous residues in the region spanning amino acids 500–1000 (Table 2). They were chosen because of their high degree of conservation among the alphaherpesvirus UL37 proteins (Appendix A). We made several AAA substitution mutants and of these, 22 that displayed restricted growth were further purified, and we confirmed the sequence of each introduced mutation in the virus genome following purification. All the mutant viruses expressed and accumulated a stable pUL37 protein in infected cells, as judged by immunoblot assays using the antibody to GFP (Figure 5A). These mutants were initially screened using low MOI infections for their ability to replicate in Vero cells (Figure 5B). Each displayed some degree of replication, thus identifying the mutated residues as non-essential sequences. To further analyze these viruses, 13 of the 22 which displayed the poorest growth properties were used to analyze single-step growth kinetics following infection of cells at an MOI of 10 PFU/cell (Figure 5C). As the data show, two mutant viruses, 657–59 and 739–41, have minimal replication over virus input and two other mutant viruses, 680–82 and 704–06, display some degree of replication but with substantial defects compared to the wild-type virus. The rest all displayed varying replication properties but sufficient to give rise to good levels of progeny virus. Comparing single- and multi-step growth of these viruses shows that, in some cases, replication is impaired over multiple rounds involving cell-to-cell spread, but significantly less so for one round of replication. This is especially evident for the mutants 672–74, 676–78, 702–04, and 713–15 (Figure 5D).

### 3.6. pUL37 Residues 657–59, 680–82, 704–06, and 739–41 are Important for Virus Morphogenesis

We further examined the four mutants that had the greatest defects in virus replication using similar TEM and confocal analyses. TEM was again carried out in infected Vero cells (Figure 6). The mutant-infected cells showed an accumulation of capsids in the nucleus (n) (black arrowheads) and electron-dense DNA-filled capsids in the cytoplasm (white arrowheads). Perinuclear enveloped capsids were observed in some infected cells (pe). Occasionally, a mature particle was observed leaving the cell (arrow) (Figure 6). RPE-1 cells were infected with EGFP-tagged pUL37 mutants and analyzed by confocal microscopy. The pUL37 alanine mutants 657–59, 680–82, 704–06, and 739–41 trafficked to a juxtanuclear site (Figure 7). In fact, the accumulation at this site was more pronounced than that for the wild-type protein, suggesting that these proteins traffic to the Golgi but the mutation abrogates virus envelopment and release from this compartment.

## 4. Discussion

Initial envelopment of the virion takes place at the inner nuclear membrane (INM). The interacting proteins, pUL31 and pUL34, the latter a membrane protein, are required for this initial envelopment; reviewed in [6,9,78,79,80]. After the capsid is enveloped at the INM, it fuses with the outer nuclear membrane (ONM), depositing a naked (non-enveloped) particle into the cytoplasm [81]. These capsids are transported to the trans-Golgi compartment (TGN) or other cytoplasmic organelle (late endosomes) for final envelopment [82,83,84]. This cytoplasmic site must accumulate all the different tegument proteins that are incorporated into the mature virion [4], and also, the lipid membrane that envelopes this particle has to contain the full repertoire of viral glycoproteins, reviewed in [5,7,8,9,10]. One of the most intriguing aspects of this morphogenesis pathway is the role of the tegument proteins in this dual envelopment process, the cellular localization and movement of tegument proteins prior to their incorporation into the maturing virus, and the viral factors/signals that traffic particles to the maturation compartment. What is still unclear is the composition of the tegument as the virus is translocated from the nucleus to the cell surface. The multitude of tegument proteins have different locations within the cell; some are exclusively cytoplasmic and others exclusively nuclear, and yet others that are detected in both compartments. Thus, as the virus particle progresses on its way to the surface, all of these tegument components must be incorporated into the final mature virion. The mechanisms of how the tegument proteins function in trafficking and virion morphogenesis, and the manner by which protein–protein interactions determine the fate of virus particle formation, are still unclear [10]. In fact, the tegument proteins appear to be required for the transition of the capsids from the site of assembly to the cytoplasmic site for final envelopment.

The trafficking of pUL37 in infected cells has been visually detected in living cells using EGFP. Pronounced accumulation was seen in the Golgi or Golgi-derived structure. This localization was coincident with that of Giantin, a Golgi marker [35]. In our previous studies, we showed that Golgi localization of pUL37 was dependent on the presence of functional pUL36 but did not require capsids [35]. This localization likely occurs at sites of virus cytoplasmic budding, and thus pUL37 localization at this site prepares for envelopment of the capsid. That said, Pasdeloup et al. [85] have shown that in the absence of pUL37, capsids do not transport to the TGN. Here we have shown that the N-terminal 480 amino acids are not required for virus production in epithelial cell lines. However, recent studies have identified domains present in the N-terminal half of the PRV and HSV pUL37 that are important mediators of axonal transport [47,49]. Thus, the N-terminus of pUL37 is an important pathogenic determinant. What is curious in our analysis of the N-terminal truncation mutants is that the ∆72N up to ∆372N are all lethal, but the virus specifying ∆481N regains the ability to replicate. In each case, productive replication corresponds to cytoplasmic virion envelopment, suggesting either the mutant polypeptides directly inhibit this maturation step, or perhaps another defect in pUL37 function leads to the phenotype. The distribution of pUL37–EGFP also changes as sequential deletions are made. The strong juxtanuclear fluorescence becomes more dispersed into smaller aggregates, of which some are located at perinuclear sites. It is possible that these sites of pUL37–EGFP accumulation are devoid of other virion components that are required for secondary envelopment and that somehow, in the ∆481N mutant, some of the relocalized pUL37–EGFP sites coincide with the correct budding sites of the virus. It is also possible that some of the N-terminal truncations have a deleterious effect on the folding of the remaining pUL37 polypeptide or they uncover a dominant-negative domain in pUL37 that inhibits virus envelopment. Self-interaction domains of pUL37 have been mapped both at the N (1–300) and C (568–1123) terminal regions and a leucine zipper at residue 203 to 224 [71]. However, of all the mutants, only ∆990C virus was difficult to grow in the complementing cell line, BD45, producing virus stocks that were on average 10-fold lower than the other mutants. This phenotype of this particular mutant, we believe, suggests a transdominant effect of ∆990C polypeptide on the wild-type protein.

The structure of the N-terminus of HSV-1 pUL37 has been elucidated at high resolution by the Heldwein Lab [48,49,86]. They show that the N-terminus has a compact bean-shaped alpha helical structure which positions the conserved R2 surface region of domain IV and is a determinant of neuroinvasion. The Δ481N mutant we describe partially disrupts domain IV and excludes residues Q403, E452, and Q455, while maintaining residues Q511 and R515, which collectively are shown to be essential for retrograde transport in neuronal axons [47]. It is possible that while disruption of this functional motif by both our laboratory and the Smith Lab maintain replicative ability in epithelial cells, the Δ481N mutant virus may also be defective for retrograde transport and neuroinvasion; we have not examined this function.

A long time ago, we isolated an insertion mutant in pUL37, in which nine amino acids were inserted at residue 480 of the protein. This virus similarly failed to replicate robustly and, instead, formed small plaques on Vero cells. When infected cells were examined by TEM, many capsids were seen in the cytoplasm adjacent to membrane structures, but it appeared that membrane closure was absent (data not shown). The Kousoulas Lab has shown that gK interacts with pUL37 and this site at 480 may be required for this physical association during secondary envelopment [61]. In fact, tyrosine residues that are located at positions 474, 476, 477, and 480 are essential for this interaction [60]. The identification of a physical interaction between pUL37 and pUS10 highlights a potential alternate pathway for virus budding. pUS10, by virtue of a predicted interaction with gE, may bridge the tegument and the envelop [62].

The C-terminus of pUL37 specifies important functions, as judged by the lethality of polypeptide truncations. In fact, the C-terminal 133 amino acid peptide is absolutely required for virus replication. All other C-terminal truncation mutants are similarly lethal for the virus and all appear to affect secondary envelopment, as judged by TEM analysis. However, the cellular distribution of some of these polypeptides differs, as judged by GFP fluorescence. It appears that in the ∆730C infected cells, fluorescence is uniformly present in both cytoplasmic and nuclear compartments, minus the Golgi accumulation. The cells infected with ∆990C revealed greater amounts of nuclear fluorescence. These data indicate the C-terminus is likely affecting pUL37 trafficking too, either by interactions with virus or host cell components. Some of these mutant polypeptides will lack the interaction domains of pUL37 (pUL36, gK/pUL20, dystonin, or TRAF6), and this may account for the observed lethal phenotype. However, because these were truncation mutations, it is difficult to distinguish potential global effects of the mutation on pUL37 function. It has also been difficult to obtain high resolution structural information of the alphaherpesvirus pUL37 C-terminal half due to degradation of the purified full-length polypeptide in this region. However, a small angle X-ray scattering (SAXS) analysis of PRV C-terminal portion of pUL37 has yielded information on the structural properties of this region [86]. This analysis showed the C-terminal region (residues 478–919) is a flexible monomer with an elongated folded core comprised of many alpha helices and an unstructured carboxy terminus. As suggested by the authors, the flexible C-terminus, which harbors many functional domains, is ideally suited for a protein that has dynamic interactions and functions at different times during the course of the virus life cycle [86].

We have also discovered, using an alanine scanning type mutagenesis, that many residues in the C-terminal half are required for robust virus production and a few are critical for virus replication, highlighting an important role for these sequences. Some of these mutants, when analyzed using imaging methods, also abolish secondary envelopment but they are still strongly associated with a juxtanuclear structure, which is likely the Golgi structure. These residues could potentially reveal new activities of this complex polypeptide. Because none of the pUL37 viral binding partners map to these residues, it will be of interest to determine what these sequences are doing in future studies. Our goal in this proposal was to map the pUL37 domains that regulate its intracellular trafficking. We now have information on this, as well as information on the importance of the different regions and residues of this complex tegument protein.

## Figures and Tables

**Figure 1 viruses-11-00853-f001:**
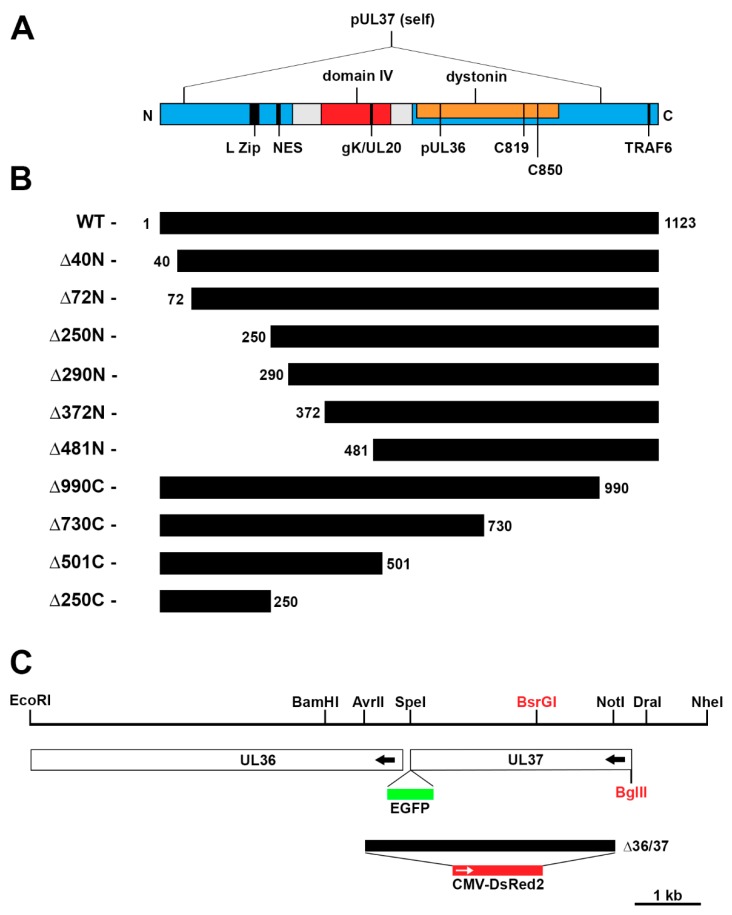
Functional and interaction domains of the HSV-1 pUL37. (**A**) The carton illustrates the 1123 amino acid linear pUL37 polypeptide. The motifs identified to act as a nuclear export signal (NES), a leucine zipper (L Zip), deamidase active sites (C819 and C850), and the virulence R2 residues in domain IV (red) are shown, as well as the self (blue), pUL36, gK/pUL20 (474–480), dystonin (orange), and TRAF6 interaction domains [38,45,66,67,71]. (**B**) Polypeptide truncations of the N- and C-terminus of pUL37 were generated molecularly and transferred into the HSV-1 genome using marker-rescue/marker-transfer methods. Mutant polypeptides expressed a C-terminal EGFP fusion. (**C**) Carton of the UL37 and UL36 genetic locus of HSV-1. The figure shows the restriction enzyme fragments used to clone different regions of UL37 and to engineer mutant genes. These include the site of the insertion of EGFP ORF and the region of a UL36–UL37 double deletion followed by insertion of the CMV-DsRed2 cassette. Engineered restriction enzyme sites are shown in red.

**Figure 2 viruses-11-00853-f002:**
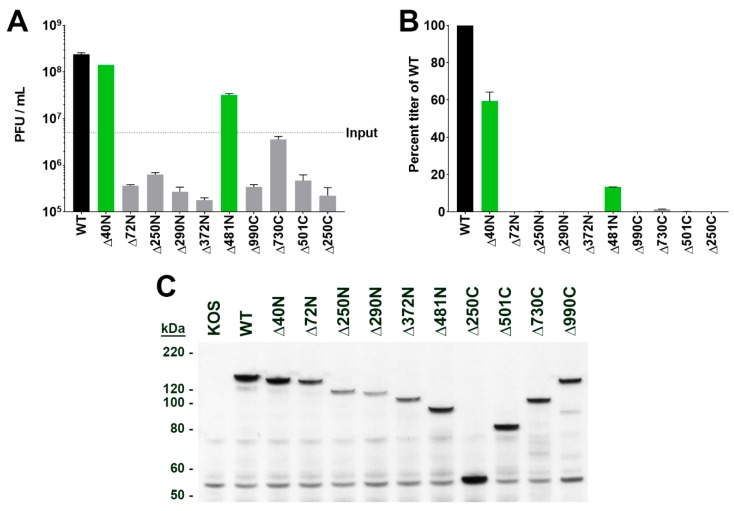
The carboxyl half of pUL37 is sufficient for replication of virus in tissue culture. Mutant viruses were analyzed by single-step growth curves. Vero cells (5 × 10^5^) were infected at MOI of 10 PFU/cell. The infected cells were harvested at 24 h post-infection. The lysates were sonicated and serial dilutions plated on BD45 cell monolayers to enumerate virus titer. Data are plotted as PFU/mL from the infected cell lysate with input virus inoculum indicated (**A**) and percent titer relative to the wild-type (WT) UL37–EGFP virus (**B**). Each value is an average of two biological replicates. (**C**) Accumulation of the pUL37 mutant polypeptides was examined by immunoblot methods. Vero cells were infected as above and total cell lysates were analyzed by SDS-PAGE (4–12% NuPAGE gradient gel), followed by transfer to nitrocellulose membranes. The blot was probed with rabbit anti-GFP antibodies to confirm stable expression of all truncated pUL37–EGFP fusion proteins. Protein standards are shown on the left of the blot.

**Figure 3 viruses-11-00853-f003:**
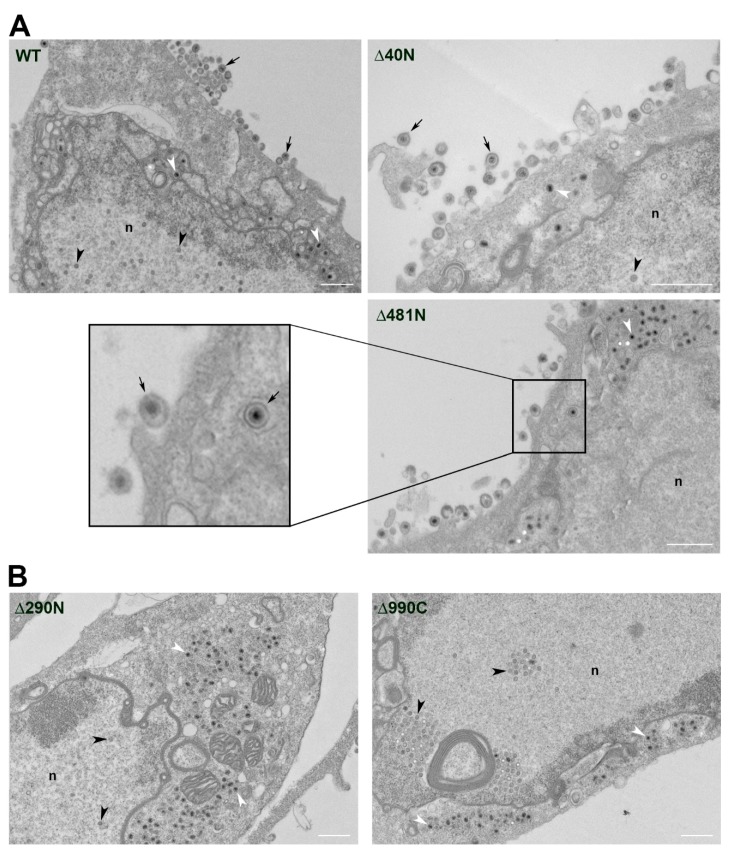
The pUL37 polypeptide sequence 481–1123 supports secondary envelopment of virus particles. Vero cells were infected with each UL37-mutant virus and analyzed by TEM at 18 h post-infection. (**A**) Particles representing normal virion morphogenesis were observed in cells infected with viruses expressing wild-type (WT), Δ40N, and Δ481N pUL37–EGFP. These include capsids in the nucleus (n) (black arrowheads), electron-dense DNA-filled capsids in the cytoplasm (white arrowheads), and enveloped virus particles in the cytoplasm and exiting the cell (arrows). (**B**) Cells shown were infected Δ290N or Δ990C pUL37–EGFP expressing viruses. Enveloped DNA-filled capsids were not observed in the cytoplasm or at the cell surface. Images are representative of all viruses which failed to produce secondary enveloped virus particles. Scale bar = 1 µm.

**Figure 4 viruses-11-00853-f004:**
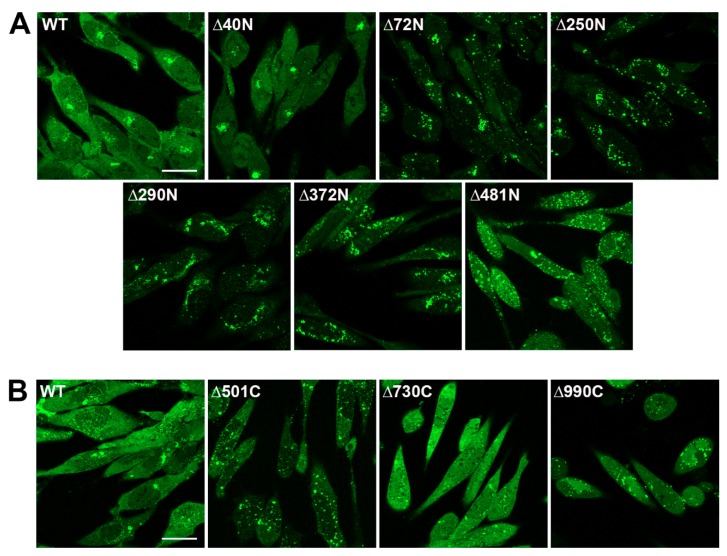
Both amino and carboxyl terminal truncations affect pUL37–EGFP trafficking. RPE-1 cell monolayers in chamber slides were infected with wild-type and each UL37–EGFP mutant virus (excluding Δ250C) at an MOI of 10 PFU/cell and analyzed by confocal microscopy 12 h post-infection. Wild-type pUL37–EGFP trafficked to the Golgi complex, as evidenced by the juxtanuclear localization and previous data from Desai et al. [35]. (**A**) N-terminal truncations of greater than 40 amino acids resulted in the dispersal of pUL37–EGFP localization to a more perinuclear localization. This was also seen with the functional Δ481N mutation. (**B**) C-terminal truncations all abolish the juxtanuclear localization pattern and they display different cellular localization, as judged by the fluorescence observed. Scale bar = 25 µm.

**Figure 5 viruses-11-00853-f005:**
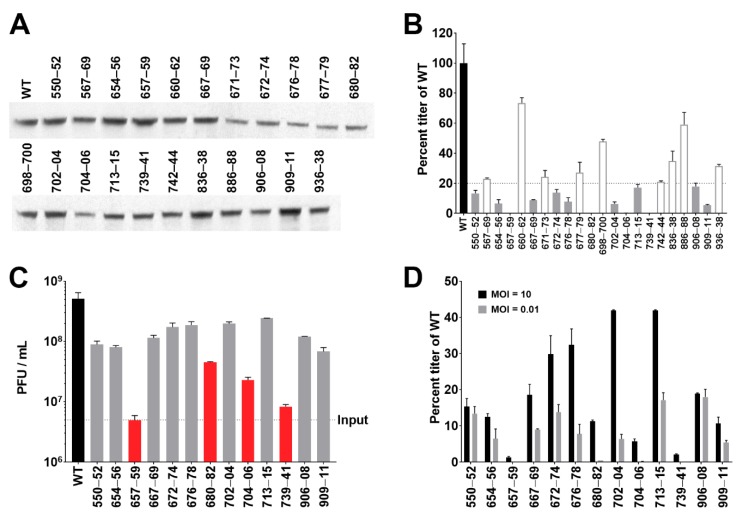
Alanine scanning mutagenesis of conserved amino acids in the C-terminus of pUL37 reveals residues important for productive replication. (**A**) Lysates from Vero cells infected with 22 triple alanine scanning mutants of UL37–EGFP and wild-type (WT) were analyzed by SDS-PAGE, as described in the legend of Figure 2, and the blot of the transferred proteins probed with anti-GFP rabbit antibodies to examine mutant pUL37–EGFP fusion expression. (**B**) Mutant viruses were analyzed for growth on Vero cells following infection of at MOI = 0.01 PFU/cell and titration of the virus burst 3 days post-infection, as described in the legend of Figure 2. Viruses with titers below 20% of WT are shaded grey, and those above 20% open bars. (**C**) The 13 viruses with most reduced titers in a multi-step growth curve were also analyzed for single-step growth by infection at MOI = 10 PFU/cell and harvested 24 h post-infection, as described in the legend of Figure 2. The data for the four most deleterious AAA mutants in the virus are shown as red bars. (**D**) Titers from the 13 viruses analyzed by both single- and multi-step growth are plotted as percent titer of WT for each corresponding experiment. All data represent the mean of at least two independent replicates with error bars of one standard deviation.

**Figure 6 viruses-11-00853-f006:**
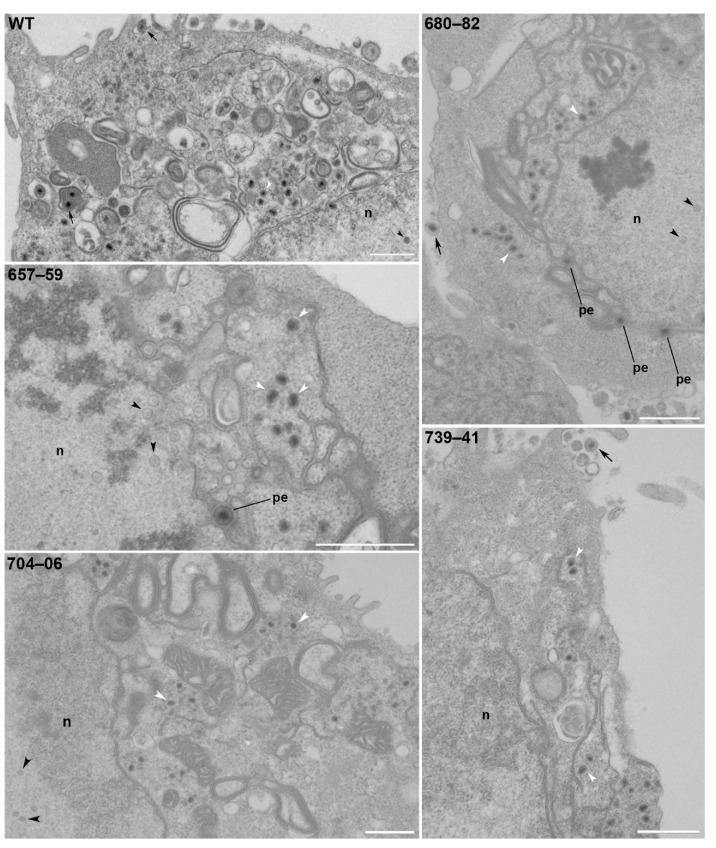
AAA substitutions at residues 657–59, 680–82, 704–06, and 739–41 of pUL37 result in significant reduction of mature enveloped virus particles. Vero cells were infected with viruses specifying AAA substitutions or WT and TEM analysis of infected cells was carried out 18 hours post-infection. In the mutant-infected cells, there was an accumulation of capsids in the nucleus (n) (black arrowheads) and electron-dense DNA-filled capsids in the cytoplasm (white arrowheads). Perinuclear enveloped capsids were observed in some infected cells (pe). Occasionally, a mature particle was observed leaving the cell infected with AAA mutant viruses (arrow). Cytoplasmic capsids surrounded by an envelope were not observed in these cells, whereas they are clearly evident in WT pUL37-infected cells. Scale bar = 1 µm.

**Figure 7 viruses-11-00853-f007:**
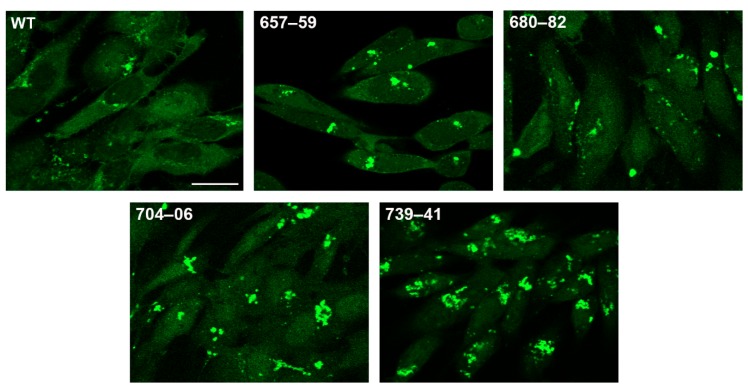
Deleterious pUL37 alanine substitution mutants accumulate GFP fluorescence at a juxtanuclear site. RPE-1 cells in chamber slides were infected with EGFP-tagged pUL37 mutants or wild-type at an MOI of 10 PFU/cell and analyzed by confocal microscopy at 12 h post-infection. The pUL37–EGFP alanine mutant polypeptides 657–59, 680–82, 704–06, and 739–41 are observed with intense fluorescent signals at a juxtanuclear site. Scale bar = 25 µm.

**Table 1 viruses-11-00853-t001:** Plating efficiency of wild-type HSV-1 strain KOS, K∆UL37, K∆UL36, and K∆36/37CR on Vero, BD45, HS30, and BH31 cells.

Virus	Vero	BD45	HS30	BH31
KOS	^a^2.28 × 10^10^	2.16 × 10^10^	3.23 × 10^10^	1.79 × 10^10^
K∆UL37	^b^NP	1.97 × 10^10^	NP	1.93 × 10^10^
K∆UL36	NP	NP	5.80 × 10^9^	7.00 × 10^9^
K∆36/37CR	NP	NP	NP	1.06 × 10^9^

^a^ Data are presented as PFU/mL. ^b^ NP: No plaque formation.

**Table 2 viruses-11-00853-t002:** UL37–EGFP triple alanine (AAA) scanning mutants.

UL37–EGFP Mutant	pUL37 Amino Acids	Mutant Amino Acids
550–52	SLL	AAA
567–69	HWD	AAA
654–56	AWA	AAA
657–59	RDF	AAA
660–62	GLG	AAA
667–69	VEG	AAA
671–73	RTK	AAA
672–74	TKL	AAA
676–78	ALI	AAA
677–79	LIT	AAA
680–82	LLE	AAA
698–700	NIE	AAA
702–04	LLR	AAA
704–06	REL	AAA
713–15	AVE	AAA
739–41	SMY	AAA
742–44	ALA	AAA
836–38	LRS	AAA
886–88	SKL	AAA
906–08	RWR	AAA
909–11	RLS	AAA
936–38	TWK	AAA

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
