# Peer review of "Functional Domains of the Herpes Simplex Virus Type 1 Tegument Protein pUL37: The Amino Terminus is Dispensable for Virus Replication in Tissue Culture"

_viruses, 2019, doi:10.3390/v11090853_

Round 1

Reviewer 1 Report

Grzesik and colleagues present a good piece of work, in which they analyzed the role of the HSV-1 inner tegument protein pUL37 via generating N- and C-terminal truncated mutants of this gene in the KOS strain using marker-rescue/marker-transfer. The authors analyzed pUL37 expression, viral growth, localization of viral proteins and envelopment of mutant viruses using Western blotting, plaque assays, fluorescence and transmission electron microscopy, respectively.

The authors show that the N-terminus, but not the C-terminus of UL37, is dispensable for virus replication in cell culture. Via alanine scanning mutagenesis, the authors identified in more detail (non-)essential domains in the C-terminus.

The careful work confirms the importance of pUL37 for the maturation process of the enveloped HSV-1. I only have two minor comments: 

In the introduction, the authors mention that the conserved R2 motif is important for neuroinvasion. From Fig. 1, it is not entirely clear whether this motif is deleted in the delta481 mutant virus? This virus strikingly behaves different compared to the other N-terminal deletion mutants in retaining some replicative capacity. The authors may comment on this point in the discussion.  Supplementary Figure 1: please indicate the mode of dilution of the viral stocks.

Author Response

Reviewer 1:

In the introduction, the authors mention that the conserved R2 motif is important for neuroinvasion. From Fig. 1, it is not entirely clear whether this motif is deleted in the delta481 mutant virus? This virus strikingly behaves different compared to the other N-terminal deletion mutants in retaining some replicative capacity. The authors may comment on this point in the discussion.

Explanation of the delta481 mutant virus and the relation of residues in the R2 motif is elaborated in the discussion section.  The aspect of neuroinvasion as it relates to this region is also addressed.  Thank you. Lines 456-461.

Supplementary Figure 1: please indicate the mode of dilution of the viral stocks.

Dilutions are 10-fold and this is now indicated in the figure legend. Thank you.

Reviewer 2 Report

Functional Domains of the Herpes Simplex Virus type 1 tegument protein pUL37:The amino terminus is dispensible for Virus Replication in Tissue Culture

This manuscript contains a comprehensive analysis of the structure and function of the HSV-1 UL37 protein achieved through the generation of mutants expressing UL37 proteins with various deletions as well as through alanine-scanning mutagenesis. The work presented clearly shows that the C-terminus of pUL37 is essential for viral replication. Importantly, the N-terminal 480 amino acids are dispensable for virus replication in tissue culture, although smaller deletions of 72 and and up to 372 amino acids were lethal. This indicates that the amino 480 aa may serve an independent functional domain that relates to neurovirulence, etc.  The triple alanine (AAA) scanning mutagenesis revealed that  mutations at 672-74,676-78,702-04 and 713-15 aa affected viral replication. Residues 657-59, 680-82,704-06 and 739-41 were important for virion morphogenesis, as these mutants trafficked to a juxtanuclear site but accumulated in these sites suggesting that while these mutants can travel to the Golgi apparatus, they negatively affect virion envelopment.

Comments:

The manuscript is written in a comprehensive and elegant manner. The authors have included a thorough review and inclusion of all relevant references including their own as the first group that performed significant work on the structure and function of the UL36/UL37 protein complex. All the results are of high quality and easily interpretable. Authors may want to increase the size of panels in figures 4, 6 and 7 for easier visualization of the results shown. Possible additions of magnified portions as insets may also help visualization as it is done in Figure 3.

Other comments:

The authors may want to comment whether amino terminal regions of 1–300 and 568–1123 of pUL37, which are involved in self-interaction prevent pUL37 from self-association causing increased binding of pUL37 to pUL36. pUL37 self-associates, when UL36 is not present or present in low amounts, suggesting that there is competition (Identification of interaction domains within the UL37 tegument protein of herpes simplex virus type Michelle A. Bucks, Michael A. Murphy Kevin J.O'Regan, Richard J.Courtney; Virology Volume 416, Issues 1–2, 20 July–1 August 2011, Pages 42-53). It would be of interest to know whether the amino terminal 481 amino acids constitute an independently functioning domain. In this regard, transient expression of the 480 amino acid domain may rescue fully the D481 truncation partial defect. The Courtney lab has also shown that the carboxy-terminal region of UL37 (residues 568–1123) partially rescues the K∆UL37 infection suggesting that the C-terminus of UL37 may contribute to its essential functional role within the virus-infected cell. This is now elegantly proven by the Desai lab, as well as revealing the specific residues required for replication and morphogenesis at the UL37 C-terminus. In future experiments, it would be of interest to look at the relative localization of the gK/UL20 complex in Δ481N mutant, as this region of pUL37 interacts with gK/UL20. It would be of interest to further elaborate on the potential role of UL37 domains and residues on retrograde and anterograde neuronal transport, since the UL37 protein interacts with microtubules and the dynein motor.

Author Response

Reviewer 2:

Comments:

The manuscript is written in a comprehensive and elegant manner. The authors have included a thorough review and inclusion of all relevant references including their own as the first group that performed significant work on the structure and function of the UL36/UL37 protein complex. All the results are of high quality and easily interpretable. Authors may want to increase the size of panels in figures 4, 6 and 7 for easier visualization of the results shown. Possible additions of magnified portions as insets may also help visualization as it is done in Figure 3.

All figures including: 4, 6, and 7 were increased in size relative to the text to make features more visible.    Thank you.

Other comments:

The authors may want to comment whether amino terminal regions of 1–300 and 568–1123 of pUL37, which are involved in self-interaction prevent pUL37 from self association causing increased binding of pUL37 to pUL36. pUL37 self-associates, when UL36 is not present or pUL37 self-associates, when UL36 is not present or present in low amounts, suggesting that there is competition (Identification of interaction domains within the UL37 tegument protein of herpes simplex virus type Michelle A. Bucks, Michael A. Murphy Kevin J.O'Regan, Richard J.Courtney; Virology Volume 416, Issues 1–2, 20 July–1 August 2011, Pages 42-53).

We cannot distinguish with our current viruses how the presence or absence of pUL36 affects the activities of the self-association domains that are encoded in the different mutant polypeptides. We would have to generate some of these mutants in a UL36 null genetic background as well as confirm the self-association domains function like those identified using transient transfection assays. Also, data from the high resolution assays show both the N-terminal half and the C-terminal half when purified in solution behave as monomers. This illustrates again the complex nature of pUL37 self-interaction. We have stated in the text “Self-interaction domains of pUL37 have been mapped both at the N (1-300) and C (568-1123) terminal regions and a leucine zipper at residue 203 to 224 [85]. However, of all the mutants only ∆990C virus was difficult to grow in the complementing cell line, BD45 producing virus stocks that were on average 10 fold lower than the other mutants. This phenotype of this particular mutant we believe suggests a transdominant effect of ∆990C polypeptide on the wild-type protein.” Lines 447-452.

It would be of interest to know whether the amino terminal 481 amino acids constitute an independently functioning domain. In this regard, transient expression of the 480 amino acid domain may rescue fully the D481 truncation partial defect. The Courtney lab has also shown that the carboxy-terminal region of UL37 (residues 568–1123) partially rescues the KΔUL37 infection suggesting that the C-terminus of UL37 may contribute to its essential functional role within the virus-infected cell. This is now elegantly proven by the Desai lab, as well as revealing the specific residues required for replication and morphogenesis at the UL37 C terminus.

This is an interesting idea and certainly is corroborated by data published. The first structural analysis of the PRV N-terminal half of pUL37 proposed this region shares similarity with cellular multisubunit tethering complexes (MTCs), which control vesicular trafficking in eukaryotic cells by tethering transport vesicles to their destination membranes. They also suggested that this region while not affecting virus production had a role in cell to cell spread. Subsequently more compelling evidence was provided for a role of the N-terminus in virus neuroinvasion. The virus that would express an N-terminal polypeptide (∆501C) cannot replicate which suggests that in our system the N-terminal 501 polypeptide cannot function independently without the essential C-terminal polypeptide. 

In future experiments, it would be of interest to look at the relative localization of the gK/UL20 complex in Δ481N mutant, as this region of pUL37 interacts with gK/UL20. It would be of interest to further elaborate on the potential role of UL37 domains and residues on retrograde and anterograde neuronal transport, since the UL37 protein interacts with microtubules and the dynein motor.

This is an important point. Because the gK/pUL20 interaction site is located in the vicinity of the 480 amino acid position future studies will address how this mutant polypeptide (∆481N) interacts with gK/pUL20.For the second part, Reviewer 1 also brought this up and we have addressed this in the discussion. Thanks